# Regulation of Epithelial and Endothelial Barriers by Molecular Chaperones

**DOI:** 10.3390/cells13050370

**Published:** 2024-02-21

**Authors:** Susana Lechuga, Armando Marino-Melendez, Nayden G. Naydenov, Atif Zafar, Manuel B. Braga-Neto, Andrei I. Ivanov

**Affiliations:** 1Department of Inflammation and Immunity, Lerner Research Institute, Cleveland Clinic Foundation, Cleveland, OH 44195, USA; lechugs@ccf.org (S.L.); marinoa4@ccf.org (A.M.-M.); naydenn@ccf.org (N.G.N.); khana28@ccf.org (A.Z.); braganm@ccf.org (M.B.B.-N.); 2Department of Gastroenterology, Hepatology and Nutrition, Digestive Disease Institute, Cleveland Clinic, Cleveland, OH 44195, USA

**Keywords:** actin cytoskeleton, adherens junctions, myosin motors, permeability, tight junctions

## Abstract

The integrity and permeability of epithelial and endothelial barriers depend on the formation of tight junctions, adherens junctions, and a junction-associated cytoskeleton. The establishment of this junction–cytoskeletal module relies on the correct folding and oligomerization of its protein components. Molecular chaperones are known regulators of protein folding and complex formation in different cellular compartments. Mammalian cells possess an elaborate chaperone network consisting of several hundred chaperones and co-chaperones. Only a small part of this network has been linked, however, to the regulation of intercellular adhesions, and the systematic analysis of chaperone functions at epithelial and endothelial barriers is lacking. This review describes the functions and mechanisms of the chaperone-assisted regulation of intercellular junctions. The major focus of this review is on heat shock protein chaperones, their co-chaperones, and chaperonins since these molecules are the focus of the majority of the articles published on the chaperone-mediated control of tissue barriers. This review discusses the roles of chaperones in the regulation of the steady-state integrity of epithelial and vascular barriers as well as the disruption of these barriers by pathogenic factors and extracellular stressors. Since cytoskeletal coupling is essential for junctional integrity and remodeling, chaperone-assisted assembly of the actomyosin cytoskeleton is also discussed.

## 1. Introduction

The formation of tissue barriers is the most important functional feature of different epithelial and endothelial cells. Epithelial barriers protect the body’s interior from environmental stress factors and pathogens, create interfaces for nutrients and waste movements, and enable the formation of different tissue compartments with unique chemical environments. Barriers created by the vascular endothelium establish a separate compartment for circulated blood and provide additional protection for sensitive internal organs, such as the brain. The integrity and permeability of epithelial and endothelial barriers depend on the assembly of intercellular junctions, which are multiprotein adhesive complexes formed at cell–cell contact sites. Tight junctions (TJs) and adherens junctions (AJs) are the most important regulators of barrier properties in various types of epithelia and endothelia [1,2,3,4,5,6,7,8]. Adhesive properties of TJs and AJs are determined by transmembrane proteins engaged in homotypic interactions with their partners on opposing plasma membranes. Members of the claudin family, occludin, and junctional adhesion molecule A are the most well-known transmembrane components of TJs, whereas classical cadherins are the major adhesive molecules in epithelial and endothelial AJs [1,2,3,4,5,6,7,8]. At the cytoplasmic face of the plasma membrane, the transmembrane TJ and AJ proteins interact with numerous scaffolding, cytoskeletal, and signaling molecules that are essential for junctional assembly and regulation. These cytoplasmic scaffolding proteins include members of a “zonula occludens” (ZO) family at TJs and catenins at AJs [1,2,3,4,5,6,7,8]. Multiple interactions between the different transmembrane and cytoplasmic TJ/AJ proteins result in the assembly of supramolecular platforms at intercellular contacts that control tissue barrier integrity and permeability.

One of the most prominent features of epithelial and endothelial AJs and TJs is their coupling with the actomyosin cytoskeleton. In well-differentiated epithelia, both junctional complexes are attached to a circumferential actin filament belt enriched by a cytoskeletal motor protein, non-muscle myosin II (NM II) [9,10]. This contractile actomyosin structure is critical for stabilizing epithelial junctions and driving their remodeling during normal tissue morphogenesis as well as in diseases associated with the disruption of tissue barriers [6,11,12]. Furthermore, coupling to the perijunctional actomyosin enables AJs and TJs to sense and transduce mechanical forces [11,12,13]. Likewise, endothelial junctions are linked to contractile cortical actomyosin bundles that regulate fluid flow and leukocyte trafficking across the vascular wall [14]. In addition to their association with the actomyosin cytoskeleton, AJs and TJs also interact with cortical microtubules. The mechanisms and functional roles of such interactions, however, remain incompletely understood [15] and will not be discussed in this review.

The junction–cytoskeleton modules are characterized by the complex architecture originating from the precise ordered assembly of their structural subunits. For example, protein interactome analysis and super-resolution microscopy have revealed the existence of different structural subdomains within epithelial junctional complexes that contain distinct molecular networks and have different dynamics during junctional remodeling [16,17]. Moreover, the perijunctional actomyosin belt in well-differentiated epithelia shows a highly ordered periodic distribution of NM II clusters that resembles the sarcomeric organization of actomyosin units in muscle cells [9,18]. While high-order organization of junctional complexes and a junction-associated actomyosin cytoskeleton is essential for the fine-tuned regulation of tissue barriers and mechanosensitive signaling at intercellular contacts, little is known about the mechanisms regulating the nanoscale organization of AJs and TJs as well as the myofibril-like assembly of contractile NM II units in the perijunctional actomyosin belt. Actin filament polymerization and myofibril assembly are generally known to be orchestrated with the folding of actin and the NM II motor domain, respectively, and require assistance from cytosolic chaperones [19,20,21]. Furthermore, published studies have revealed the association of several AJ and TJ proteins with various molecular chaperones [22,23,24,25]. The specific roles of chaperones in regulating tissue barrier integrity and junctional assembly, however, remain poorly understood.

In this review, we discuss the accumulating data regarding the roles of molecular chaperones in regulating intercellular junctions and the barrier properties of epithelial and endothelial layers in vitro and in vivo. We will primarily focus on cytoplasmic chaperones and will pay less attention to endoplasmic reticulum (ER) and mitochondrial chaperones involved in co-translational protein folding and protein quality control. Furthermore, we will not discuss cellular responses and signaling pathways specifically stimulated by secreted extracellular chaperones. The major focus of this review is on the regulation of epithelial and endothelial junctions, and in addition to that, we will also discuss important chaperones and co-chaperones that control the assembly of the actomyosin cytoskeleton, even if their role in regulating intercellular junctions and tissue barriers have yet to be determined.

## 2. Heat Shock Proteins as a Key Part of the Cellular Chaperome

Cell and tissue homeostasis is regulated by an elaborate “chaperome”, representing a complex network of chaperones, co-chaperones, and other associated molecules [26,27]. This molecular network contains more than 300 different proteins that play key roles in regulating normal proteostasis and in controlling plasticity and adaptation of the cellular proteome under pathological conditions [28,29]. Heat shock proteins (HSPs) are the most studied members of the cellular chaperome, with various essential functions under homeostatic conditions, environmental stressors, and disease states [20,30,31,32,33,34]. HSPs are located in different cellular compartments where they assist in protein folding, regulate protein oligomerization and the formation of multiprotein complexes; they also prevent the aggregation of misfolded proteins and dissociate protein aggregates. Since many important HSPs are enriched in the cytoplasm, they are accessible for interactions with various cytoskeletal and adhesion proteins at the cell cortex and can regulate the assembly and dynamics of cellular adhesions and the associated cytoskeleton. The majority of studies addressing the chaperone-dependent regulation of epithelial and endothelial junctions examined HSPs and their co-chaperones; therefore, this protein network will be the focus of the present review (Figure 1).

The HSP superfamily is subdivided into seven protein families based on their molecular weight and sequence similarity [35,36]. These families are HSPA (Hsp70), HSPB (small HSPs), HSPC (Hsp90), HSPD/E (Hsp60/Hsp10), HSPH (Hsp110), DNAJ (Hsp40), and TRiC/CCT. Their first and last names represent the new and the old nomenclature systems, respectively, for these proteins, both of which are widely used in the existing literature. HSPD and TRiC/CCT proteins are also called chaperonins because they self-assemble into large oligomeric complexes. The major HSP families consist of many members. For example, there are 13 HSPA, 11 HSPB, and 41 DNAJ genes in the human genome [36]. The reasons for such genetic diversity of HSPs are unclear; however, different HSPs may have unique functional roles by interacting with specific populations of the client proteins. It is noteworthy that not all the listed HSP families act as bona fide chaperones capable of either the de novo folding or refolding of their clients. Some of them serve as co-chaperones that select and deliver clients to other HSPs acting as truth chaperones. Members of almost all the HSP families have been implicated in the regulation of epithelial and endothelial barriers and intercellular junctions and will be discussed in this review (Figure 1). Since virtually nothing is known about the barrier-regulating functions of DNAJ/Hsp40 proteins, these co-chaperones will not be specifically discussed. We will first discuss the two most studied HSP families, HSPA/Hsp70 and HSPC/Hsp90, along with their key co-chaperones, followed by a description of small HSPs/HSPBs and two chaperonin families.

## 3. HSPA/Hsp70 Chaperones

HSPAs, which are better known as Hsp70s, comprise a family of well-conserved versatile molecular chaperones. Humans express 13 different HSPA proteins, some of which are heat-inducible, while others are constitutively expressed [30,37,38]. HSPA chaperones have a variety of cellular functions that include de novo protein folding, regulation of polypeptide translocation across membranes, dissolution of abnormal protein aggregates, and assembly of different multiprotein complexes [38]. HSPA activity and specificity are tightly controlled by their association with many co-factors. The most important ones are co-chaperones of the J-domain protein (JPD) family (also known as DNaJ and Hsp40) and nucleotide exchange factors that control protein client binding and release by HSPAs [37,38]. Interestingly, recent profiling of the Hsp70 interactome revealed critical roles of JPD co-chaperones in determining the specificity of Hsp70 interactions with their clients [39]. The mechanisms of Hsp70–client interactions are extensively studied and have been reviewed elsewhere [30,37,38]. Here, we briefly outline the major steps of the chaperone cycle. The Hsp70 molecule is composed of the N-terminal domain that binds ATP and the C-terminal domain that interacts with unfolded client proteins. The folding cycle starts with the JPD co-chaperone binding to a specific client, followed by their interactions with the ATP-loaded Hsp70. The client transiently binds to an open C-terminal peptide-binding site of Hsp70. Consequently, these interactions trigger ATP hydrolysis that, in turn, results in conformational changes in the Hsp70 molecule, stabilizing its interactions with the client and causing release of the JPD co-chaperone. Next, the Hsp70–client complex binds to a nucleotide exchange factor that facilitates the exchange of Hsp70-bound ADP for ATP, resulting in the release of the folded client from the chaperone. If necessary, the released clients can further refold spontaneously or be transferred to other chaperones or chaperonin complexes for the ultimate folding and maturation. The described process highlights the complexity and tight regulation of the chaperone activities of HSPA/Hsp70s [30,37,38].

A number of recent studies suggest that several members of the HSPA/Hsp70 family act as positive regulators of epithelial and endothelial junctions and tissue barriers in vitro and in vivo (Table 1 and Figure 2). For example, siRNA-mediated knockdown of HSPA1A/Hsp72 in tumor-derived human A549 lung epithelial cells and MCF7 mammary epithelial cells markedly decreased E-cadherin protein expression and disrupted AJ assembly, resulting in an altered cell shape and increased motility [40]. Similarly, depletion of another Hsp70 chaperone, HSPA8/Hsc70, in NRK-52E normal rat kidney epithelial cells decreased E-cadherin levels and diminished the assembly of E-cadherin-based AJs [41]. Consistent with the reported stabilizing effects of Hsp70 proteins on normal AJs, overexpression or pharmacological upregulation of these chaperones by geranylgeranylacetone (GGA) reversed the loss of E-cadherin expression and AJ disassembly in NRK-52E and A549 epithelial cells during the transforming growth factor (TGF)-β-induced epithelial to mesenchymal transition (EMT) [42,43].

Interestingly, some HSPA/Hsp70 chaperones appear to be dispensable for normal TJ assembly and barrier establishment, as was shown in a model intestinal epithelium. Indeed, an antisense knockdown of HSPA1A/Hsp72 in human colon cancer-derived Caco-2 epithelial cells affected neither the transepithelial electrical resistance (TEER) nor the mannitol flux, which is indicative of an unaltered paracellular permeability for ions and small uncharged molecules, respectively [44,45]. By contrast, protective roles of HSPAs in the injured intestinal epithelium have been reported by several studies (Figure 2). For example, HSPA1A/Hsp72 depletion in Caco-2 cells exaggerated barrier disruption and disorganization of the perijunctional actin cytoskeleton triggered by either an oxidative agent, monochloramine [44], or *Clostridium difficile* toxin A [45]. Furthermore, overexpression of Hsp72 in Caco-2 cells attenuated AJ disassembly caused by gliadin, a peptide linked to intestinal barrier dysfunction in celiac disease patients [46].

The barrier-protective activity of HSPA/Hsp70 chaperones has also been described in vascular endothelial monolayers in vitro (Table 1). A recently discovered member of this protein family, HSPA12B, was found to be enriched in endothelial cells and induced by proinflammatory stimuli such as bacterial lipopolysaccharide (LPS) [47,48]. Overexpression of HSPA12B in human umbilical vein endothelial cells (HUVEC) attenuated an LPS-induced increase in endothelial permeability, according to TEER and FITC–dextran flux assays [47,48]. This barrier-protective effect of HSPA12B overexpression was paralleled by the restoration of VE-cadherin expression and increased levels of myosin light chains [47]. Similar endothelial-protective activity was described in experiments with a chemical inducer of Hsp70 chaperones, TRC051384 [49]. This Hsp70 inducer reversed the increased permeability and restored the diminished expression of VE-cadherin, occludin, and ZO-1 in LPS-activated HUVEC monolayers.

Importantly, the tissue-barrier-protective functions of HSPA/Hsp70 proteins have been also documented in several animal models of tissue injury and inflammation (Table 1 and Figure 2). For example, a combined knockout of HSPA1A/Hsp72 and HSPA2/Hsp70-3 isoforms in mice markedly increased colonic mucosal injury and inflammation induced by either dextran sodium sulfate (DSS) administration or adaptive T-cell transfer [25,51]. Consistently, intestinal epithelial-specific overexpression of Hsp70 attenuated colonic injury and inflammation in DSS-treated animals [51]. Hsp70-null mice demonstrated a more pronounced disruption of the gut barrier integrity and colonic epithelial TJ disassembly during DSS colitis compared with wild-type controls [25], suggesting that stabilization of the gut barrier is a part of the mucosal protective functions of Hsp70 chaperones. In addition to the intestinal epithelium, protective effects of Hsp70s have been observed in other tissue barriers in vivo. For example, pharmacological induction of HSPA/Hsp70 family members by GGA attenuated renal tubular injury in the unilateral urethra obstruction model in rats [42]. Furthermore, such Hsp70 induction normalized AJ integrity and restored E-cadherin expression in the injured renal epithelium [42]. Transgenic mice with overexpression of HSPA12B were protected from myocardial ischemia/reperfusion injury, and this protection was accompanied by decreased vascular permeability and the enhancement of endothelial TJs [50]. Consistently, intratracheal administration of HSPA12B siRNA that depleted expression of this chaperone in the airways increased vascular leakage in the septic lungs [47].

The molecular mechanisms underlying the described barrier-protective effects of HSPA/Hsp70 chaperones remain poorly characterized. One possible mechanism may involve the direct stabilization of junctional proteins and the assisted assembly of AJ and TJ complexes. Indeed, Hsp70 was shown to colocalize with TJs in mouse small-intestinal and colonic epithelia in vivo [25,52]. Furthermore, the physical association between Hsp70 and ZO-1 was detected by immunoprecipitation analysis in both the intestinal epithelium and cerebral vascular endothelium [25,53]. A detailed analysis is lacking, however, for the interactions between HSPA/Hsp70 chaperones and AJ/TJ proteins. An alternative mechanism of the barrier-protective effects of HSPA/Hsp70s may involve the regulation of the actomyosin cytoskeleton (Figure 3). For example, overexpression of Hsp70 that attenuated a stimuli-induced disruption of the intestinal epithelial barrier also inhibited disassembly of the perijunctional actin cytoskeleton [44,45], suggesting that these events could be causally connected. Little is known about the mechanism by which HSPA/Hsp70 chaperones regulate actin filaments and cytoskeletal motors in epithelial or endothelial cells. Interestingly, two recent reports suggested the involvement of RhoA, a small GTPase, which functions as a master regulator of the actomyosin cytoskeleton. Depletion of HSPA1A/Hsp72 in rat proximal tubular renal epithelial cells prevented the inactivation of RhoA and the disassembly of actin stress fibers triggered by pharmacological inhibition of the angiotensin II receptor [54]. This study suggests that Hsp72 serves as a positive regulator of epithelial RhoA activity by a yet to-be-determined mechanism. In a different study, however, conducted in HeLa human cervical cancer cells, Hsp70 was shown to mediate a local inactivation of AJ-associated RhoA [55]. Such inactivation involved the Hsp70-assisted assembly of a multiprotein complex containing a ubiquitin receptor, Fas-associated protein 1, and a negative regulator of RhoA activity, IQGAP1 [55]. The described opposite regulation of RhoA activity by Hsp70s is not surprising given the large number of clients and signaling pathways controlled by these chaperones. Their regulation of RhoA activity is likely to be tissue- and stimuli-dependent. More work is needed to elucidate how members of the HSPA/Hsp70 family control the assembly and function of the actomyosin cytoskeleton associated with epithelial and endothelial junctions.

## 4. HSPH/Hsp110 Chaperones

This small family of heat shock proteins with four human members shares high homology with HSPA/Hsp70 chaperones and are frequently described together as parts of a bigger superfamily [36,56]. While HSPHs can interact with unfolded proteins, they alone do not induce protein folding due to weak ATPase activity [30,36]. Instead, these chaperones serve as exchange factors for Hsp70s by accelerating the transition from the ADP-bound to the ATP-bound form in the functional cycle of these chaperones [57]. Although HSPH family members are increasingly recognized as important pro-oncogenic molecules [56], their roles in regulating epithelial barriers remain poorly understood. The most known example of such regulation was revealed when HSPH2 (also known as HSPA4, or Apg-2) was identified as a binding partner for a key TJ scaffold, ZO-1, in MDCK kidney epithelial cells [24] (Figure 2). HSPH2 was found to partially localize at TJs, with its junctional localization and interaction with ZO-1 being enhanced by heat shock [24]. Downregulation of HSPH2 expression, while not preventing TJ assembly and establishment of the paracellular barrier, did significantly slow down the rate of junctional assembly [58]. HSPH2 depletion resulted in other interesting phenotypes, including the inhibition of 3-D cyst formation and blockage of the G1/S phase transition in the cell cycle [24,58]. The latter phenotype was linked to the altered activity of a specific transcription factor, ZONAB, that competes with HSPH2 for ZO-1 binding [24]. Since major phenotypes of HSPH2 depletion were also recapitulated by ZO-1 knockdown, it is thought that this chaperone modulates junctional assembly and epithelial morphogenesis by controlling ZO-1 folding and function [58].

Another line of evidence implicating the HSPH/Hsp110 chaperones in the regulation of epithelial barriers emerged from studying kidney function in mice with a total knockout of the HSPH3 gene (also known as HSPA4L or Apg-1). In the kidneys, HSPH3 was shown to be selectively expressed in epithelial cells of the cortical segment of the distal tubule [59]. Interestingly, homozygous HSPH3 knockout mice developed kidney hydronephrosis and a loss of osmotolerance [59]. Although the mechanisms underlying such abnormal renal functions have not been investigated, disruption of kidney epithelial barriers could be an important contributor to this pathology. Overall, the existing evidence suggests that members of the HSPH/Hsp110 family could play essential roles in regulating epithelial barrier integrity in vivo and in vitro. This evidence, however, is limited only to the renal epithelium, and it remains unknown if barrier-promoting functions of HSPHs are kidney-specific or could also be important for other epithelial or endothelial barriers.

## 5. HSPC/Hsp90 Chaperones

Members of the HSPC family are critical chaperones that interact with various clients and co-chaperones, thereby having versatile cellular functions. In mammals, this protein family consists of five members. Best known are the heat-inducible Hsp90α (also named HSPC1 or HSP90AA1) and constitutively expressed Hsp90β (HSPC3 or HSP90AB1) that are predominantly located in the cytoplasm along with the endoplasmic reticulum glucose-regulated protein 94 (GRP94; also known as HSPC4) and the mitochondrial TRAP1/HSPC5 [60,61]. HSPC/Hsp90s are classical “foldases” that ensure the correct folding of a large number of clients. According to published estimations, cytoplasmic Hsp90 paralogs participate in the folding of approximately 10% of the eucaryotic proteome [62]. Their interactome consists of more than 2000 proteins, many of which play essential roles in regulating transcription, DNA repair, the cell cycle, apoptosis, cell differentiation, and lipid and carbohydrate metabolism, among many other key cellular functions [63].

The minimal functional unit of Hsp90 is a dimer, which can further homo- or hetero-oligomerize with co-chaperones to create large multimeric complexes [61]. The Hsp90 monomer consists of three domains: an N-terminal domain with an ATP-binding site, a middle domain that interacts with clients, and a C-terminal domain responsible for Hsp90 dimerization [60,61]. All three domains have docking sites for different Hsp90 co-chaperones that regulate the activity cycle of Hsp90 and provide specificity for its interactions with different clients. ATP binding to Hsp90 initiates a series of conformational transitions that lead to chaperone dimerization and interactions with clients and co-chaperones. The Hsp90 chaperone cycle has been extensively studied and is described in detail in recent reviews [60,61,64].

The published Hsp90α/Hsp90β interactome includes several important AJ and cytoskeletal proteins, such as E-cadherin, β-catenin, NM IIA, RhoA, and actin-capping proteins [63]. It is expected, therefore, that Hsp90s promote the establishment of epithelial and/or endothelial barriers and the assembly of intercellular junctions. Paradoxically, a large body of evidence indicates the opposite, where the activity of HSPC/Hsp90 chaperones has negative effects on tissue barrier integrity and can drive their disruption under inflammatory, ischemic, and other pathological conditions (Table 2; Figure 2). This evidence originates from studies utilizing different pharmacological inhibitors of HSPC/Hsp90 chaperones. The barrier-protective effects of Hsp90 inhibitors have been most extensively documented in the activated vascular endothelium in vitro and in vivo. For example, the Hsp90 inhibitors such as radicicol and 17-(Allylamino)-17-demethoxygeldanamycin (17-AAG) attenuated LPS-induced disruption of the paracellular barrier in bovine pulmonary arterial endothelial cells (BPAEC) [65] and human lung microvascular endothelial cells (HLMVEC) [66]. Consistent with this, Hsp90 inhibition prevented the LPS-induced disassembly of endothelial AJs and rescued the decrease in VE-cadherin expression [65]. Significant barrier-protective effects of Hsp90 inhibitors were documented in BPAEC and HLMVEC activated by a variety of stimuli, including TGF-β, thrombin, phorbol ester [67], hydrochloric acid [68], and the SARS-CoV2 spike protein [69].

A common mechanism of endothelial barrier disruption by different external stimuli involves remodeling of the perijunctional actomyosin cytoskeleton. A key feature of such cytoskeletal remodeling is the stimulation of actomyosin contractility driven by the small GTPase, RhoA. Interestingly, Hsp90 inhibitors prevented remodeling of the actomyosin cytoskeleton in activated endothelial cells by decreasing RhoA and NM II functions [66,67,68,69] (Figure 3). Little is known about the mechanisms underlying the Hsp90-dependent regulation of RhoA signaling. Since RhoA belongs to the group of Hsp90 clients, it can be directly folded and activated by these chaperones. Alternatively, Hsp90s can activate this GTPase indirectly by targeting upstream regulators such as guanine nucleotide exchange factors [82]. It is noteworthy that HSPC/Hsp90s were shown to regulate other signaling cascades in activated endothelial cells that could act either upstream or in parallel to RhoA to induce endothelial barrier disruption. These signaling events include the activation of NF-κB [69,83], Src kinase [65], AKT, and MAP kinases [68,84]. Another mechanism underlying the barrier-protective effects of Hsp90 inhibitors could involve the activation and nuclear translocation of heat shock factor 1, which stimulates the expression of HSPA/Hsp70 chaperones [85,86]. This mechanism is in line with known barrier-protective functions of Hsp70s (Table 1), but its roles in the stabilization of epithelial and vascular barriers by Hsp90 inhibitors have not been tested.

Consistent with the described in vitro studies, pharmacological inhibitors of HSPC/Hsp90 chaperones were shown to protect vascular endothelial barriers in vivo (Table 2). For example, 17-AAG treatment attenuated endotoxin-induced vascular leakage in rat retinas [70]. Another Hsp90 inhibitor, 17-dimethylaminoethylamino-17-demethoxygeldanamycin (17-DMAG), blocked the disruption of the blood–brain barrier in a rat model of intracerebral hemorrhage [71], mouse models of cerebral ischemic stroke [72], and traumatic brain injury [73]. In these animal models, the attenuation of vascular permeability caused by Hsp90 inhibition was accompanied by the restored expression of key TJ proteins, such as occludin, ZO-1, and claudin-5 [70,71,72,73]. Similar to the described in vitro studies, in vivo protection of the vascular barriers is not a specific effect of Hsp90 inhibitors. Rather, it is a part of their general anti-inflammatory and tissue-protective activities that result in suppressed NF-kB signaling, decreased cytokine production, and upregulation of AKT activity [70,71,72,73].

The junction-enhancing effects of Hsp90 inhibitors are not limited to the vascular endothelium and were also observed in different epithelial cells (Table 2). For example, Hsp90 inhibition by 17-AAG promoted AJ assembly in human corneal epithelial cells [74], whereas other Hsp90 inhibitors, geldanamycin and ganetespib, increased the expression and junctional accumulation of E-cadherin in human colonic adenocarcinoma cells [75,76]. Furthermore, geldanamycin treatment attenuated the increased barrier permeability and TJ disassembly in MDCK cells triggered by the activation of a Gα12 subunit of heteromeric G proteins [77]. The mechanisms that could mediate such detrimental effects of HSPC/Hsp90s on epithelial junctions remain poorly investigated. During Gα12 activation in MDCK cells, Hsp90 was shown to link together Gα12 and the Src tyrosine kinase, leading to increased Src activity. Activated Src phosphorylated ZO-1 and ZO-2, which resulted in TJ disassembly [77]. In human colonic adenocarcinoma cells, Hsp90 interacts with a key ubiquitin ligase, Hakai, and protects it from proteasomal degradation [75]. Such Hakai stabilization could disrupt epithelial junctions by accelerating the ubiquitination and degradation of different AJ and TJ proteins.

## 6. GRP94/HSPC4 Chaperone

Glucose-regulated protein 94 (GRP94), also known as glycoprotein 96 (Gp96) or HSPC4, is an ER-resident member of the HSPC family that is essential for protein folding, secretion, and quality control in the ER [87,88]. While the primary focus of this review is on non-ER chaperones with cytoskeletal/junctional clients, we decided to discuss GRP94 due to its unique cellular dynamics and unconventional functions at the cell cortex. One of the remarkable features of GRP94 is its stimuli-induced translocation from the ER to the plasma membrane [89]. This translocation is most commonly characterized under inflammatory conditions, and it is essential for the GRP94-dependent regulation of mucosal inflammation and host–pathogen interactions. For example, GRP94, while being barely detectable at the enterocyte plasma membrane in normal intestinal mucosa, was found to be enriched at the apical surface of intestinal epithelial cells in Crohn’s disease patients [90]. In the inflamed intestinal epithelium, GRP94 colocalized with an apical plasma membrane protein, CEACAM6, a known receptor for adherent-invasive *Escherichia coli* (AIEC), a Crohn’s disease-related pathogen [91]. Furthermore, GRP94 was shown to be essential for AIEC invasion in intestinal epithelial cells [91]. Other functions of plasma-membrane-localized GRP94 in regulating host–pathogen interactions include binding to *Clostridium difficile* Toxin A and the *Listeria monocytogenes* virulence protein, Vip, in intestinal epithelial cells [92,93], as well as mediating influenza A virus interactions with alveolar epithelial cells [94]. Interestingly, plasma membrane-translocated GRP94 has a unique conformation due to its altered N-glycosylation, and it was shown to form multiprotein complexes with different receptors and adhesion proteins at the cell surface [95].

Another interesting activity of GRP94 translocated to the plasma membrane involves the remodeling of the cortical actomyosin cytoskeleton, as described in cultured HeLa cells exposed to the *Listeria monocytogenes* pore-forming toxin, listeriolysin O [78]. The toxin was shown to promote GRP94 binding to NM IIA, leading to the assembly of the chaperone and myosin-containing plasma membrane blebs [78]. Such GRP94-induced remodeling of the cortical actomyosin cytoskeleton serves as a protective cellular response against bacteria-induced membrane rupture. Regulatory activity of GRP94 toward the actomyosin cytoskeleton was also observed in epithelial cells without toxin treatment, where GRP94 depletion promoted activation of NM II by the phosphorylation and assembly of actomyosin stress fibers [78] (Table 2).

Given the plasma membrane localization of GRP94 in inflamed epithelia and its roles in the remodeling of the cortical actomyosin cytoskeleton, one can suggest that this chaperone can regulate epithelial barriers during tissue inflammation and infection. Such barrier-modulating activities of GRP94, however, remain poorly investigated. A *Drosophila* homolog of this chaperone, Gp93, is known to be a critical regulator of intestinal homeostasis that controls nutrient absorption, growth factor signaling, and tissue growth [79]. A Gp93 mutation that markedly reduces its expression in *Drosophila* tissue resulted in the defective secretion of gastric acid and the diminished absorption of different amino acids in the fly’s gut. These defects in ionic/nutrient fluxes could be indicative of a distortion of the apico-basal polarization of Gp93-deficient epithelial cells, leading to the abnormal localization of membrane ion channels and transporters. Indeed, loss of Gp93 altered the architecture of the apical plasma membrane domain and caused the abnormal morphology of septate junctions, which are the *Drosophila* analogs of mammalian TJs [79].

Consistent with the *Drosophila* study, deletion of GRP94 in mice resulted in altered tissue homeostasis, especially in rapidly self-rejuvenating tissues such as the intestine [80,96] (Table 2). Specifically, loss of this chaperone caused the rapid erosion of the small intestinal epithelium due to the inhibition of stem-cell renewal and abnormal Wnt and Notch signaling. This resulted in a profound disruption of the gut barrier and bacterial translocation into the liver and lymph nodes [80]. Interestingly, GRP94 knockout triggered a selective disruption of TJs in the mouse ileum without noticeable detrimental effects on AJ integrity [80]. Loss of GRP94 expression was also shown to affect intercellular adhesions in the liver [81]. Specifically, a selective deletion of GRP94 in hepatocytes impaired the formation of several junctional complexes, manifested by the decreased membrane accumulation of E-cadherin and the impaired assembly of connexin channels at gap junctions (Figure 2). Furthermore, loss of GRP94 in hepatocytes induced a hyperproliferative response and oncogenic signaling [81,97], although it remains unknown whether junctional disruption contributes to such increased cell proliferation.

## 7. STIP1/HOP Protein

Stress-inducible phosphoprotein 1 (STIP1), also known as Hsp70-Hsp90 organizing protein (HOP), is an important co-chaperone that regulates the key chaperone cycle involving Hsp70 and Hsp90 [98,99]. Despite binding ATP and having ATPase activity, STIP1/HOP cannot fold protein substrates alone and, therefore, itself is not a molecular chaperone. The overall structure of this protein is well conserved in yeasts, mice, and humans and contains several alpha-helical domains organized in a tandem fashion. Specifically, STIP1/HOP has three tetratricopeptide repeats (TPR), called TPR1, TPR2A, and TPR2B, along with two aspartate–proline (DP) motifs, which mediate its interactions with HSPs and other proteins [98,99]. STIP1/HOP simultaneously binds the Hsp70 and Hsp90 dimer via its TPR1 and TPR2A domains, respectively. Formation of this Hsp70-HOP-Hsp90 complex is critical for the activity of the Hsp70-Hsp90 chaperone cycle and correct folding of client proteins. While Hsp70 alone can efficiently fold some substrates, it arrests the folding of other clients due to the highly hydrophobic nature of its substrate-binding chamber [98,100]. To complete their folding, these clients must be transferred from Hsp70 to Hsp90, which highlights the orchestrated activity of these two essential cytoplasmic chaperones. STIP1/HOP plays a key role in this cycle by bringing together two chaperones and accelerating the client transfers from Hsp70 to Hsp90 [98,99]. Furthermore, STIP1/HOP inhibits the ATPase activity of HSP90, thereby keeping Hsp90 in an open conformation that is accessible to its protein clients.

An interesting feature of STIP1/HOP is the prominent role of this co-chaperone in regulating the architecture of the actin cytoskeleton. STIP1/HOP directly interacts with actin via the TRP2A domain [101] (Figure 3), and depletion of STIP1/HOP was shown to disrupt the organization of actin filaments in renal epithelial and breast cancer cells [101,102,103]. The underlying molecular mechanisms may not depend on direct actin folding but rather on the regulation of actin-binding and actin-signaling molecules. For example, STIP1/HOP depletion increased expression of the actin-depolymerizing protein, cofilin-1, while downregulating levels of profilin, which is essential for actin filament polymerization [101]. Furthermore, STIP1/HOP can regulate actin filament dynamics and architecture by affecting the expression and activity of small GTPases such as RhoC and Rnd1 [102,103]. Consistent with its role in organizing the actin cytoskeleton, STIP1/HOP plays a positive role in regulating ECM adhesion and the migration of cancer and endothelial cells [99,103,104,105].

Surprisingly, little attention has been paid to the possibility of STIP1/HOP-dependent regulation of epithelial or endothelial barriers. One study reported that this co-chaperone binds to a known AJ and polarity protein, Scribble, in MDCK cells [106] (Figure 2A). STIP1/HOP colocalized with Scribble at epithelial junctions and regulated Scribble stability; however, loss of neither STIP1/HOP nor Scribble affected AJ integrity or the formation of polarized epithelial cysts [106]. This thereby suggests the dispensability of this co-chaperone for normal epithelial morphogenesis. Consistently, STI1, a *C. elegans* homolog of this co-chaperone was found to be highly expressed in the intestine. STI1-null mutants, however, did not display prominent gastrointestinal abnormalities, suggesting an unaltered integrity of the gut barrier [107]. By contrast, STIP1 deficiency in mice resulted in embryonic lethality, accompanied by the failure of neural tube closure and increased cell apoptosis in STIP1-null embryos [108]. Overall, while known STIP1/HOP roles in regulating the Hsp70-Hsp90 chaperone cycle and the actin cytoskeletal assembly allude to the possible involvement of this co-chaperone in intercellular junction and tissue barrier assembly, these possible functions of STIP1/HOP await testing in future studies.

## 8. NUDC Co-Chaperones

Nuclear distribution C (NUDC) is an important family of molecular co-chaperones with a major function in coordinating the chaperone activity of Hsp70 and Hsp90 [109]. While mechanisms of their actions are just beginning to emerge, NUDC co-chaperones are thought to accelerate the client transfer from Hsp70 to Hsp90, bypassing the canonical HOP pathway [109]. Four members of this family are termed NUDC, NuDCD1 (also known as CML66), NUDCD2 (NUDCL2), and NUDCD3 (NUDCL) [110], with each family member displaying specificity to different domains in their protein clients. For example, NUDC selectively binds to WD40 repeats, NuDCD1 interacts with RNA helicases and the COPI complex, and NUDCD2 associates with RCC1 domains, whereas NUDCD3 binds to Kelch domains [111]. These different binding preferences are likely to be important for several cellular activities of the NUDS family members. While most of the known functions of NUDS proteins are related to the regulation of the microtubules and a microtubule motor, dynein [110], accumulating evidence suggests that these co-chaperones also control the architecture of the actomyosin cytoskeleton and the integrity of tissue barriers.

NUDC, a founding member of this protein family, is highly expressed in different epithelia, including the intestinal mucosa, fallopian tubes, and bronchial epithelium [112]. Recent studies demonstrated that loss of this co-chaperone disrupted the organization of the actin cytoskeleton in retinal pigment epithelial cells [113] and renal carcinoma cells [114]. Interestingly, the mechanistic link between NUDC and the actin cytoskeleton in both experimental systems involves a key actin-depolymerizing protein, cofilin-1 (Figure 3). NUDC was shown to bind cofilin-1 and protect it from proteasomal degradation [113,114]. Given the important role of cofilin-1 in regulating epithelial barrier integrity [115], one could suggest that NUDC plays barrier-protective roles in epithelial layers. Little evidence is available to support such a role, although NUDC depletion in zebrafish resulted in hydrocephalus and pronephric duct dilation, which are the phenotypes usually associated with disruption of epithelial barriers [113].

A different mechanism targeting the actin cytoskeleton has been attributed to NUDCL2. This co-chaperone was found to interact with NM IIA in cervical and lung cancer cells [116] (Figure 3). Furthermore, NUDCL2 depletion diminished protein expression of NM IIA by promoting its proteasomal degradation. NUDCL2 regulates NM IIA stability in an Hsp90-dependent fashion, since Hsp90 overexpression restored NM IIA protein levels in NUDCL2-deficient epithelial cells [116]. While it remains unknown whether NUDCL2 controls the integrity of epithelial barriers, such a possibility is supported by recent clinical data. Specifically, rare biallelic variants of the NUDCL2 gene have been shown to markedly reduce the tissue expression of this protein [117]. Such downregulation of NUDCL2 expression resulted in multiple organ pathologies including epithelia-related dysfunctions, hepatic cholestasis, and chronic renal failure [117]. This clinical data suggests that NUDCL2 could play important roles in regulating epithelial barrier integrity and function in different organs.

Unlike other members of the NUDC protein family, NUDCL1 was shown to affect epithelial cell adhesion by regulating AJ protein expression. This regulatory mechanism was demonstrated in colonic, pancreatic, and lung cancer-derived epithelial cells characterized by the increased abundance of this co-chaperone [118,119,120]. Downregulation of NUDCL1 expression in different cancer cell lines resulted in a marked increase in E-cadherin protein levels. This effect was indicative of a canonic EMT since it was accompanied by decreased expression of the mesenchymal markers N-cadherin and vimentin [118,119,120]. The antagonizing effect of NUDCL1 on E-cadherin protein expression is inconsistent with the possibility that E-cadherin serves as a client for NUDCL1-assisted folding. Instead, NUDCL1 likely regulates E-cadherin expression indirectly, by modulating intracellular signaling events. Indeed, in colonic and lung cancer cells, NUDCL1 was found to act as a positive regulator of the insulin growth factor receptor–ERK signaling pathway, which is essential for defining the epithelial cell phenotype and plasticity [118,119].

## 9. UNC-45A

UNC-45A is an important co-chaperone that assists in the folding of a subpopulation of Hsp90 clients [111]. There are two reasons for highlighting the functions of this co-chaperone in the present review. The first is its role in regulating the folding and activity of key cytoskeletal motors, myosins [121,122], and the second is the existence of human genetic syndromes caused by UNC-45A mutations that are characterized by the marked dysfunction of epithelial barriers (Table 3). This co-chaperone belongs to the evolutionarily conserved UCS (UNC-45/Cro1/She4p) protein family, which, in vertebrates, has two different paralogs, UNC-45A and UNC-45B. UNC-45A is ubiquitously expressed in different tissues while UNC-45B is mainly expressed in muscle cells [123]. UNC-45A contains three different domains: the N-terminal tetratricopeptide (TPR) domain, the central region, and the C-terminal USC domain [121,122,124]. The USC domain binds and folds conventional myosin II motors as well as some unconventional class I and V myosins [121,125,126,127,128,129]. The TPR domain interacts with microtubules and destabilizes the microtubule structure [130,131,132]. Studies in *C. elegans* revealed a functionally important oligomerization of UNC-45 mediated by intermolecular interactions between the TPR and USC domains [121,127]. Such oligomerization creates linear molecular arrays of this co-chaperone that assist in the assembly of folded myosin II monomers and dimers into highly ordered myofilaments (Figure 3).

Several genetic variants of UNC-45A have been linked to rare genetic disorders such as the osteo-oto-hepato-enteric syndrome, the Aagenaes Syndrome, and the congenital diarrheal disorder [133,134,135,136,137] (Table 3). The disease-causing mutations are located in different regions of the UNC-45A gene, ranging from the 5′ untranslated region to the C-terminal UCS domain [133,134,135,136,137]. Many of these mutations decrease the protein level of UNC-45A in patients’ cells, indicating that the loss of UNC-45A expression and function is a causal driver of these diseases [133,136,137]. The major symptoms of UNC-45A-linked disorders, such as diarrhea, hepatic cholestasis, and hearing loss, strongly suggest an abnormal epithelial homeostasis and impaired functions of epithelial barriers. This is supported by the ultrastructural examination of the intestinal epithelium of patients with congenital diarrhea that revealed defective microvilli and the abnormal localization of apical plasma membrane transporters [137].

Important roles of UNC-45A in regulating epithelial barriers and homeostasis were further elaborated using model organisms such as zebrafish and *Drosophila* as well as in human epithelial cell lines (Table 3). An unc-45a zebrafish mutant recapitulated the intestinal epithelial abnormalities observed in human patients, which include decreased microvilli length and density and the appearance of intracellular vacuoles containing apical plasma membrane proteins [137]. Furthermore, UNC-45A-deficient zebrafish mutants failed to develop intestinal crypt-like structures, indicating an abnormal development of the crypt-villous axis [133]. The only study that investigated the effect of UNC-45 depletion of the gut barrier integrity in vivo was performed in *Drosophila* mutants, where decreased UNC-45 expression resulted in increased gut permeability in aging flies [138].

Cellular mechanisms underlying the abnormal epithelial homeostasis caused by UNC-45A depletion have been recently investigated in human intestinal epithelial cells in vitro, which revealed the complexity of this co-chaperone’s function [129,137,138]. One study utilizing SK-CO15 and HT-29 human colon cancer-derived epithelial cells observed that UNC-45A knockout caused a marked disruption of the model intestinal epithelial barrier manifested by increased paracellular permeability and AJ/TJ disassembly [138]. Consistent with this, assembly of the perijunctional actomyosin belt and the contractility of actomyosin bundles at the apical plasma membrane were diminished in the UNC-45A-depleted cells. The described UNC-45A-dependent regulation of barrier integrity appeared to involve myosin binding but was independent of its interactions with microtubules. This was concluded based on the selective dominant negative effects of overexpression of the C-terminal UCS-domain-containing part of the UNC-45A molecule that selectively binds to the myosin motor but not microtubules [138]. Two other studies using Caco-2 human colonic epithelial cells reported additional functional effects of the UNC-45A deletion [129,137]. These effects included defects in the apico-basal cell polarity, abnormal vesicle trafficking at the apical plasma membrane, and disorganization of apical microvilli, which closely recapitulate the abnormal phenotypes of UNC-45A-depleted enterocytes of human patients [137]. Overall, the described clinical and experimental studies highlight UNC-45A as a critical regulator of the epithelial barrier, intercellular junctions, and cell polarity, at least in the intestinal epithelium. Given that UNC-45A functions as an Hsp90 co-chaperone, one could suggest that its functions at epithelial barriers are orchestrated by Hsp90 or other major chaperones; however, the mechanisms regulating UNC-45A activity in vertebrate epithelia remain to be investigated.

## 10. HSPB (Small Heat Shock Proteins)

HSPBs or small heat shock proteins represent a subfamily of molecular chaperones widely expressed in animals, plants, bacteria, and some viruses [139,140]. In mammals, there are 10 different members of this subfamily (HSPB1 through to HSPB10) that are characterized by either ubiquitous (HSPB1, HSPB5, HSPB6, and HSPB8) or tissue-specific expression [139,141]. Interestingly, only HSPB1, HSPB5, and HSPB8 appear to be inducible by high temperatures and other environmental stressors [142]. HSPBs are relatively small proteins with molecular weights in the range of 15 to 40 kDa. The characteristic structural signature of these chaperones is a highly conserved alpha-crystallin domain in the central part of the molecule, which is flanked by structurally variable N-terminal and C-terminal domains [30,141]. The N-terminal and central domains of the HSPB molecules play major roles in their interactions with different clients [30,141]. HSPBs have a high propensity for self-association, and they primarily exist in different oligomeric states, ranging from dimers to large multimeric complexes composed of 20–40 subunits [141,142]. These oligomers are highly dynamic and assemble in either a homotypic fashion or contain different members of the HSPB family.

Unlike other HSP chaperones, HSPBs do not bind ATP and cannot refold client proteins. Instead, they act as “holdases” by interacting with partially unfolded proteins and protecting them from aggregation [143,144]. These partially unfolded and stabilized clients can then be transferred by HSPBs to true foldases such as Hsp70 for their refolding (Figure 1). The anti-aggregation activities of HSPBs most likely underline the cytoprotective functions of these chaperones. Several members of the HSPB family are known to inhibit major cell death pathways, such as apoptosis and ferroptosis, and attenuate the cellular senescence induced by various extracellular stressors [139,141,142]. The mechanisms underlying the cytoprotective activities of HSPBs are likely to be complex since these chaperones interact with many client proteins, including crucial kinases, cytoskeletal components, apoptotic proteins, ubiquitin ligases, and inflammatory factors [140,142]. Importantly, the cell-protective activities of HSPBs can be modulated by their posttranslational modifications, most notably by phosphorylation. HSPB1 is phosphorylated at three serine residues (Ser15, Ser178, and Ser82) in its N-terminal domain, primarily by mitogen-activated protein-kinase-associated protein (MAPKAP) kinases 2 and 3 [145,146]. MAPKAP kinases are downstream effectors of the well-known stress-sensitive p38 kinase, whose activation plays a major role in the phosphorylation of HSPB chaperones in stressed cells. In addition to the p38-MAPKAP cascade, other important protein kinases such as PI3K/AKT, ERK1/2, PKC, PKD, and cAMP/PKA have been shown to phosphorylate different HSPBs [139,145]. Phosphorylation is known to modulate HSPB oligomerization and activity. Specifically, it induces the disassembly of large protein oligomers into smaller species [147,148,149]. The effects of the phosphorylation-induced HSPB depolymerization on their functions are likely to be context-dependent since both activation and inhibition of the chaperone activity of HSPBs by their phosphorylation were reported [148,149,150].

Although the effects of HSPBs on the integrity and stability of epithelial and endothelial junctions have been extensively investigated in vitro and in vivo, the majority of these studies were focused on a single member of this family, HSPB1, also known as Hsp27 in humans and Hsp25 in mice (Table 4). Hsp27 was found to be enriched at lateral intercellular contacts in normal endothelial cells [151] and either in heat-stressed or ATP-depleted renal epithelial cells [152]. Such localization suggests that HSPB1/Hsp27 can regulate intercellular adhesions and tissue barriers. An important epithelial junction-dependent mechanism, which is controlled by Hsp27, is the EMT of epithelial-derived cancer cells. Hsp27 is commonly upregulated in different tumors [153,154], and its effects on tumor cell behavior have been investigated by using knockdown and overexpression approaches. Downregulation of HSPB1/Hsp27 expression by RNA interference increased the expression of a key AJ protein, E-cadherin, in colorectal, pancreatic ductal, and prostate cancer cells [155,156,157]. In addition, the loss of Hsp27 promoted AJ and TJ assembly in prostate cancer cells [157]. Consistent with this, Hsp27 overexpression decreased E-cadherin protein levels and triggered junctional disassembly in different cancer cells [155,156,157]. This data suggests that Hsp27 negatively regulates AJ integrity in different epithelia-derived human tumor cells. The only study not consistent with such a conclusion was performed in normal rat kidney proximal epithelial cells (NRK-52E), where the overexpression of human HSPB1/Hsp27 increased steady-state E-cadherin protein levels and attenuated the loss of E-cadherin during TGFβ-induced EMT [158]. The reasons for such conflicting data remain unclear since Hsp27-dependent regulation of steady-state junctions in normal non-transformed epithelia is poorly investigated. One could suggest, however, that tumorigenic cell transformation could switch HSPB1/Hsp27 functions in epithelial cells from a barrier-protective to a barrier-destabilizing mode.

A large body of evidence demonstrated the protective roles of HSPB1/Hsp27 in stressed epithelial and endothelial cells (Table 4; Figure 2B). For example, ATP depletion of human or mouse renal tubular epithelial cells resulted in a rapid disruption of the paracellular barrier and AJ disassembly, accompanied by a dramatic disorganization of the actin cytoskeleton [159]. Overexpression of human Hsp27 attenuated the epithelial barrier disruption, AJ disassembly, and cytoskeletal disorganization in metabolically stressed renal epithelial cells, whereas siRNA-mediated depletion of Hsp27 exaggerated these events [159]. Additionally, overexpression of Hsp27 in Caco-2 human colonic epithelial cells protected the paracellular barrier of these cells from the disruption caused by exposure to *Clostridium difficile* toxin B [160].

Consistent with results obtained in model epithelial monolayers, HSPB1/Hsp27 was shown to protect the integrity of endothelial barriers in a phosphorylation-dependent fashion. Overexpression of this chaperone in human brain microvascular endothelial cell (HBMEC) monolayers significantly reduced the permeability triggered by oxygen and glucose deprivation [161]. Furthermore, Hsp27 overexpression inhibited AJ and TJ disassembly and cytoplasmic redistribution of junctional proteins in the metabolically stressed HBMEC. Interestingly, oxygen and glucose deprivation caused the rapid actin polymerization and assembly of actin stress fibers in endothelial cells, and these cytoskeletal alterations were also inhibited by Hsp27 overexpression [161]. Hsp27 knockdown did not affect the steady-state permeability of normal human dermal microvascular endothelial cells (HDMEC) [163]. Loss of Hsp27 expression, however, significantly exaggerated the disruption and attenuated recovery of the endothelial barrier during HDMEC activation by thrombin. Similarly, enhancement of the barrier-disruptive effects of thrombin was observed after the exposure of endothelial monolayers to a J2 compound, which specifically inhibits Hsp27 oligomerization [163]. Together, these data suggest that HSPB1/Hsp27 could stabilize endothelial junctions and protect the barrier integrity of stressed and activated endothelial monolayers. Because Hsp27 phosphorylation is a common consequence of cellular activation and stresses, its effects on endothelial barrier integrity were also investigated. Overexpression of human Hsp27 with triple mutations mimicking its phosphorylation (S15D, S78D, and S82D) in rat pulmonary microvascular endothelial cells markedly suppressed the increased endothelial permeability and the appearance of intercellular gaps triggered by either hypoxia or TGFβ treatment [162]. Interestingly, the introduction of this Hsp27 phosphomimic resulted in the formation of actin stress fibers, which appeared to be essential for barrier stabilization in Hsp27-overexpressed endothelial cells [162]. In a different study, a triple mutant of Hsp27 that cannot be phosphorylated (S15A, S78A, and S82A) showed a decreased ability to attenuate a thrombin-induced endothelial barrier disruption in HUVEC cells compared with wild-type Hsp27 [163]. Together, these data suggest that phosphorylation is essential for the barrier-protective effects of Hsp27 in endothelial cells.

Consistent with its in vitro functions, HSPB1/Hsp27 has been shown to protect the integrity of vascular endothelial and epithelial barriers in vivo (Table 4). Many of these studies utilized transgenic mice with global overexpression of human HSPB1/Hsp27 and examined the experimentally induced tissue ischemia in different organs. For example, hepatic ischemia and reperfusion injury resulted in increased cell death and inflammation in both liver and kidney tissues and increased Evans-blue extravasation in these organs, which is indicative of increased leakage of the renal and hepatic vasculature [164,165]. Hsp27-overexpressing mice showed a significant reduction in tissue damage, inflammation, and vascular leakage in the kidneys and the liver after hepatic ischemia/reperfusion [164,165]. These tissue-protective effects of Hsp27 were associated with the preservation of the actin cytoskeletal integrity. Indeed, ischemia/reperfusion was accompanied by decreased labeling intensity of apical F-actin in proximal tubular epithelial cells as well as basolateral F-actin in hepatocytes and bile canaliculi of control mice, which suggests depolymerization of the actin cytoskeleton. Such ischemia/reperfusion-driven cytoskeletal disruption was significantly attenuated in the kidneys and the liver of Hsp27-overexpressing animals [164,165]. Similar findings were reported in a brain ischemia/reperfusion model that resulted in a marked breakdown of the blood–brain barrier, causing water and blood plasma proteins to leak into the brain parenchyma [166]. This was paralleled by the decreased expression of endothelial junctional proteins such as VE-cadherin, occludin, and ZO-1. Overexpression of human Hsp27 significantly attenuated vascular permeability and restored junctional expression in the ischemic brain [166]. Interestingly, cell-intrinsic protective effects of Hsp27 were observed in mice with the specific overexpression of HSPB1/Hsp27 in the vascular endothelium [161]. In the above study, cerebral ischemia/reperfusion induced the disruption of the blood–brain barrier, the redistribution of junctional proteins from the plasma membrane of brain microvascular endothelial cells, and the induction of endothelial actin stress fibers in control mice [161]. Remarkably, all these structural and functional abnormalities of the ischemic brain vessels were significantly attenuated in mice with endothelial overexpression of Hsp27 [161]. Much less is known about the protective effects of HSPB1/Hsp27 on epithelial barriers during tissue injury and inflammation. Transgenic mice with selective overexpression of human Hsp27 in the kidney proximal tubules were shown to be resistant to renal fibrosis caused by a unilateral urethral obstruction [167]. These effects were associated with AJ stabilization and inhibited the cytosolic translocation of E-cadherin and β-catenin in tubular epithelial cells [167]. Together, the described studies implicate Hsp27 in protecting vascular endothelial barriers during tissue ischemia in vivo by stabilizing endothelial junctions and regulating the remodeling of the endothelial actin cytoskeleton.

In contrast to the significant focus on HSPB1/Hsp27, only a few studies have investigated the roles of other HSPB family members in the regulation of epithelial or endothelial barriers (Table 4). For example, HSPB5 (also known as αB-crystallin) was shown to interact with E-cadherin/β-catenin complexes in a HONE1-2 human nasopharyngeal carcinoma cell line [168] (Figure 2A) as well as selectively associate with cadherin-16 in human kidneys [175]. Overexpression of HSPB5 in HONE1-2 cells upregulated E-cadherin expression and triggered AJ assembly [168]. In these cells, HSPB5 was shown to stabilize AJs via binding to the E-cadherin–catenin complexes and inhibiting E-cadherin internalization [168]. Surprisingly, the opposite effects of HSPB5 on epithelial AJs were reported in human colon cancer cells, where the depletion of HSPB5 increased E-cadherin expression [169,170]. This effect of HSPB5 depletion was part of general alterations in the cancer cell phenotype, reflecting suppression of the EMT [170]. It is presently unknown whether conflicting regulatory effects of HSPB5 on the AJ assembly are due to cell-specific responses or yet-to-be-defined differences in the experimental setups. To our knowledge, only one recent study has demonstrated barrier-protective activity of HSPB5 in vivo [171]. In this study, intraperitoneal injections of cell-permeable recombinant HSPB5 reversed the increase in gut barrier permeability caused by DSS colitis in mice. Such barrier protection was accompanied by the increased expression of ZO-1 and E-cadherin in the colonic epithelium of HSPB5-injected animals. It should be noted that the described effects of HSPB5 were not specific for gut barrier integrity but were part of the general anti-inflammatory and pro-cell-survival activities of the administered chaperone [171].

Protective effects of another small heat shock protein, HSPB8/Hsp22, in the ischemic vascular endothelium have been recently reported in vitro and in vivo. Overexpression of HSPB8 in a bEnd.3 endothelial cell line did not affect the steady-state monolayer permeability [172]. In cell monolayers subjected to oxygen/glucose deprivation and reperfusion, however, HSPB8 overexpression significantly attenuated the breakdown of the endothelial barrier based on TEER and FITC-dextran flux data [172]. This in vitro ischemia/reperfusion was accompanied by increased actin polymerization and the assembly of stress fibers in bEnd.3 cells. The described alterations in the actin cytoskeleton were also inhibited by HSPB8 overexpression [172]. In agreement with the in vitro data, overexpression of HSPB8 in the mouse brain significantly attenuated the disruption of the blood–brain barrier caused by either cerebral ischemia and reperfusion [173] or intracerebral hemorrhage [174]. These protective effects of HSPB8 overexpression were accompanied by the restoration of the expression of key endothelial TJ proteins, claudin-5 and occludin [173,174]. It remains to be investigated whether remodeling of the actin cytoskeleton mediates the vascular barrier-protective activity of HSPB8 in the brain vasculature as suggested by the in vitro study [172].

Molecular mechanisms underlying the described effects of HSPB chaperones on epithelial and endothelial junctions remain poorly investigated and likely involve the regulation of different cellular structures and processes. Functions of HSPBs can be divided into three broad categories: prevention of protein aggregation, regulation of the redox status of the cell, and controlling the cytoskeletal architecture and dynamics [139,140,141]. While no direct evidence demonstrates that the first two action modes are essential for HSPB-dependent tissue barrier regulation, accumulating data support the importance of actin cytoskeletal remodeling. Indeed, the founding member of the HSPB subfamily, HSPB1/Hsp27, was initially characterized as an actin-binding protein [176], and several other HSPBs (HSPB5, HSPB6, HSPB7, and HSPB8) were shown to interact with and regulate actin filaments [177,178,179,180] (Figure 3). The mechanisms and functional consequences of such interactions remain the subjects of ongoing debates. Studies in cell-free systems demonstrated that Hsp27 selectively caps the fast-growing “plus” end of the actin filament, thereby preventing their polymerization [176]. This actin-capping was exclusively attributed to the non-phosphorylated, monomeric form of the chaperone [181]. The actin-capping activity suggests that Hsp27 should promote actin filament depolymerization. Overexpression of HSPB1/Hsp27 in different cells, however, resulted in opposite effects: acceleration of actin polymerization and protection of actin filaments from depolymerization caused by either cytochalasin D or environmental stressors [182,183,184]. Furthermore, Hsp27 phosphorylation was shown to be essential for its ability to regulate actin filament stability and remodeling [183,184,185]. The described studies can be reconciled by suggesting two different association modes for Hsp27 and actin filaments with different functional consequences [179]. One mode is a destabilizing capping of actin filaments by unphosphorylated Hsp27 monomers, whereas another mode involves stabilizing interactions of the phosphorylated Hsp27 with the lateral sides of actin filaments (Figure 3).

HSPB1/Hsp27 actions at epithelial and endothelial junctions are consistent with the second mode of actin filament binding. Hsp27 was shown to stabilize actin filaments in epithelial and endothelial cells and attenuate actin filament disassembly by barrier-disrupting stimuli (Table 4). Furthermore, at least in endothelial cells, Hsp27 phosphorylation was shown to be essential for the stabilization of intercellular junctions and the actin cytoskeleton [162,163]. In addition to a direct association with actin filaments, HSPBs can regulate the integrity and dynamics of the actin cytoskeleton indirectly by controlling RhoA signaling (Figure 3). For example, HSPB1/Hsp27 was shown to inhibit RhoA activity in neurons [186], whereas HSPB8 inactivated RhoA in endothelial cells and myocytes [172,177]. How HSPBs inhibit RhoA activity remains poorly investigated. The inhibitory effect of HSPB1 was linked to the downregulation of an upstream Rho activator, PDZ-RhoGEF [186], whereas HSPB8 appears to suppress RhoA-ROCK signaling by stimulating autophagy [172]. It remains to be investigated if the inhibition of RhoA signaling is essential for the HSPB-dependent regulation of epithelial and endothelial junctions.

## 11. HSPD/Hsp60 Chaperonins

HSPD, which is also known as a chaperonin family, is a highly conserved protein family initially described in prokaryotes, mitochondria, and chloroplasts [187]. The most-studied members of this family include the bacterial GroEL and eukaryotic Hsp60 proteins. HSPD chaperonins assemble large oligomeric complexes composed of two attached heptameric rings and a co-chaperonin ring positioned on the top of the chaperonin structure [188]. Similarly to other HSPs, they assist in the folding of a large number of nascent polypeptides and refold misfolded proteins in an ATP-dependent fashion. HSPD proteins have diverse cellular functions, some of which could be independent of protein folding. These functions include the regulation of cell survival and apoptosis, signaling, the immune response, tissue repair, and epithelial morphogenesis [30,189,190].

In *Drosophila* eggs, Hsp60 was found to accumulate at the cell–cell contact area of follicular and nurse cells, where it colocalized with the cortical F-actin belt [191]. A Hsp60 mutation that abrogated its mRNA and protein expression resulted in the marked disorganization of cellular layers and defects in egg-chamber development. These defects were accompanied by an abnormal AJ assembly and the decreased accumulation of prejunctional actin filaments [191]. Hsp60 was previously shown to fold β-actin [192], whereas its ability to interact with junctional proteins remains unknown (Figure 3). It is likely, therefore, that this chaperonin can regulate epithelial junctions by controlling the assembly of the perijunctional F-actin belt. In addition to *Drosophila* oogenesis, Hsp60 was shown to be essential for epithelial differentiation in the mammalian gut. Inducible deletion of the HSPD1 gene either in the intestinal epithelium or intestinal stem cells resulted in a loss of cell stemness, decreased cell proliferation, and abnormal Paneth cell development [193,194]. These abnormalities were linked to mitochondrial dysfunction in epithelial cells, which is consistent with the known localization of mammalian Hsp60 in mitochondria and its involvement in mitochondrial functions [190]. Loss of intestinal epithelial HSPD1/Hsp60 resulted in a low level of mucosal inflammation, which could be indicative of gut barrier disruption [193]; however, neither barrier permeability nor the integrity of intestinal epithelial junctions were investigated in Hsp60-knockout animals.

Non-canonical mechanisms of the Hsp60-dependent disruption of epithelial barriers were discovered in studies of pathogen interactions with human epithelial cells. For example, Hsp60 was shown to translocate to the surface of intestinal epithelial cells, where it interacted with a Listeria adhesion protein (LAP) and mediated *Listeria* attachment to epithelial cells [195,196]. Furthermore, LAP-Hsp60 interactions were essential for *Listeria* to induce inflammatory signaling, barrier disruption, and junctional disassembly in cultured colonic epithelial cells [195]. Interestingly, the barrier-disrupting effects of LAP were linked to increased myosin light-chain kinase expression and the activation of actomyosin contractility; however, the exact role of Hsp60 in the stimulation of actomyosin functions has not been elucidated [195].

A different barrier-disrupting signaling mechanism mediated by HSPD1/Hsp60 has been recently identified in *Streptococcus*-infected brain microvascular endothelial cells [197,198]. The *Streptococcus* infection triggered HSPD1 translocation from the mitochondrial membrane into the cytoplasm, resulting in the disruption of the endothelial barrier due to apoptotic cell death [198]. Such barrier-disrupting and pro-apoptotic activities of HSPD1 appear to be dependent on this chaperonin’s interactions with β-actin, leading to the decreased expression of an anti-apoptotic X-linked inhibitor of apoptosis [198]. β-actin is a known regulator of gene expression and cell survival [199,200]; however, it remains unknown how altered HSPD1 interactions with β-actin determine the proapoptotic function of this chaperonin.

## 12. TRiC/CCT Chaperonins

A tailless complex polypeptide 1 ring complex (TRiC), also known as chaperonin containing tailless complex polypeptide 1 (CCT), is a prominent eukaryotic cytoplasmic chaperonin that has several essential cellular functions [201,202,203]. TRiC/CCT consists of two oligomeric rings, each composed of eight homologous subunits, termed CCT1-5 in lower eukaryotes and CCTα-θ in mammals [201,202,203]. These oligomeric rings associate with each other, forming a central cavity, which serves as a place for protein client folding [204,205]. The TRiC/CCT interactome includes a large number of polypeptides with diverse structures and functions [206]. The most abundant molecular clients of TRiC/CCT, however, are the cytoskeletal proteins, such as actin and tubulin [204,205,207]. Recent structural analysis revealed that actin monomers are unable to fold spontaneously and become trapped in an unstable intermediate step [204]. Binding to TRiC/CCT is critical for transforming such a frustrated intermediate into a native actin fold [204]. These structural data are consistent with studies in cultured cells where the depletion of individual CCT subunits resulted in the marked disorganization of actin filaments [208,209].

Despite playing critical roles in regulating actin folding and cytoskeletal architecture, little is known about the roles of the TRiC/CCT chaperonin in the regulation of epithelial or endothelial barriers. Loss of the CCT5 subunit in *Caenorhabditis elegans* resulted in a profound disorganization of the apical actin cytoskeleton, accompanied by abnormal morphology of the apical plasma membrane and decreased microvilli length [210]. Surprisingly, these defects in epithelial organization did not result in increased intestinal permeability to large molecules (Texas Red–dextran), although a detailed investigation of gut barrier structure and permeability in CCT5-null animals was not performed [210]. Likewise, depletion of the CCTη and ζ subunits inhibited actin polymerization and attenuated epithelial wound healing in *Drosophila* imaginal discs [211]. This finding demonstrates the importance of TRiC/CCT for epithelial repair; however, it remains unknown if intercellular junctions were also disrupted in the chaperonin-depleted epithelial sheets, contributing to the defective wound healing. The described *Drosophila* study implicating TRiC/CCT in the regulation of epithelial repair is in line with different reports of this chaperonin’s pro-metastatic function in cancer. Indeed, increased expression of different CCT subunits is a characteristic feature of many solid tumors, which plays a causal role in promoting tumor cell invasion and metastasis [201,212]. While nothing is known about the roles of the TRiC/CCT chaperonin in controlling the integrity and permeability of mammalian epithelial and vascular barriers, such regulatory roles were suggested by recent interactome analyses of apical junctional complexes. Specifically, two different studies focusing on intracellular binding partners for the key TJ proteins claudins and Par3/Pals1 in MDCK cells revealed their interactions with seven different CCT subunits [22,23] (Figure 2A). Furthermore, the interaction between claudin-3 and this chaperonin in intact epithelial cells was confirmed by the proximity ligation assay [22]. Overall, these data suggest a possible dual mechanism for TRiC/CCT-dependent regulation of epithelial barriers that involves the formation of claudin-based TJ strands and the assembly of the junction-associated actin cytoskeleton.

## 13. Conclusions

The integrity and dynamics of epithelial and endothelial barriers depend on the assembly of protein megacomplexes at the plasma membrane, which encompass intercellular junctions and the junction-associated cytoskeleton. Molecular components of these complexes are either delivered from the biosynthetic pathway to the plasma membrane or are locally translated at the intercellular adhesions. Correct folding and assembly of the adhesion and cytoskeletal proteins are essential for the establishment of cell–cell junctions and barrier properties in various types of epithelia and the vascular endothelium. Studies over the last decades have revealed the assembly of epithelial and endothelial junctions, and the organization of the underlying actin cytoskeleton receives critical assistance from different chaperones and co-chaperones that localize at the cell cortex or on the plasma membrane. Most of these chaperones belong to the molecular network of HSPs with their accessory proteins. While functions of HSPs have attracted significant attention in neurogenerative diseases, aging, and cancer, our understanding of their roles in regulating epithelial and endothelial barriers remains in its infancy. Only a handful of chaperones and co-chaperones have been implicated in junctional biogenesis, leaving several important unanswered questions and unmet needs.

First, we need to build a comprehensive “junctional chaperome” and determine the interactions and functional interplay between key chaperones and chaperonins composing this molecular network. In this context, it is important to resolve a lingering discrepancy between the reported barrier-protective roles of HSPA/Hsp70s and the apparent barrier-destabilizing effects of HSPC/Hsp90s, which is inconsistent with their involvement in the common functional chaperone cycle. Another important goal will be to define unique functions of different co-chaperones at intercellular junctions since these co-chaperones could be responsible for the selective assembly of different junctional complexes or even topographically different subzones within the same junction. Furthermore, more work is needed to understand the roles and mechanisms of the chaperone-assisted organization of the perijunctional actomyosin cytoskeleton and how such cytoskeletal assembly influences the formation and remodeling of intercellular adhesions. Finally, it is essential to learn about alterations in the junctional chaperome under inflammatory conditions and during infections by different pathogens. This could lead to the discovery of novel mechanisms that regulate the disruption and repair of epithelial and endothelial barriers during tissue injury and inflammation.

## Figures and Tables

**Figure 1 cells-13-00370-f001:**
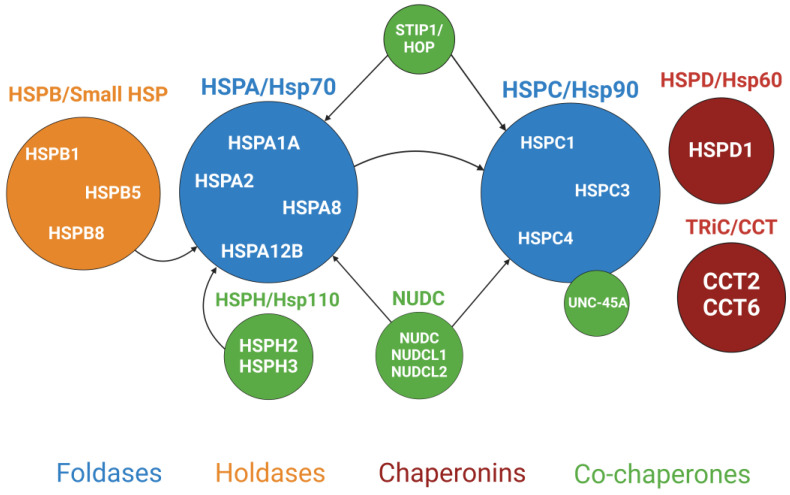
The junctional–cytoskeletal “chaperome”. The figure shows the families of chaperones, co-chaperones, and chaperonins implicated in the regulation of epithelial and endothelial junctions and/or the actomyosin cytoskeleton. The chaperone family names are depicted above the circles, whereas gene names for the individual family members that reportedly regulate intercellular junctions and the cytoskeleton are presented within the circles.

**Figure 2 cells-13-00370-f002:**
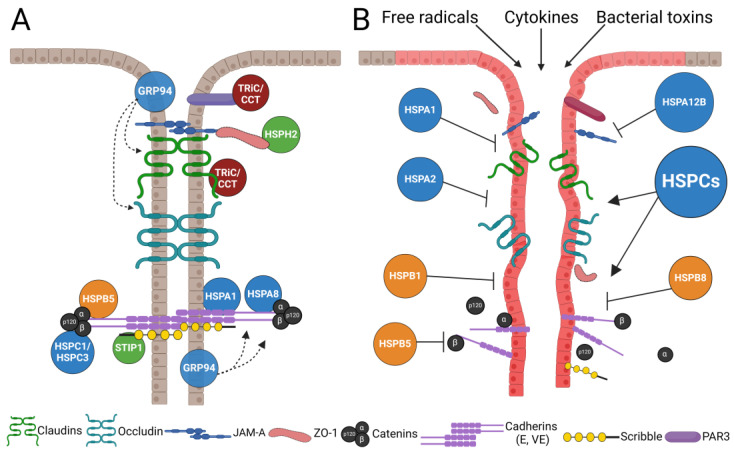
Chaperone-mediated regulation of intercellular junction integrity under homeostatic conditions and during the stimuli-induced disruption of tissue barriers. (**A**) The diagram shows the documented interactions of different chaperones and co-chaperones with AJ and TJ proteins in normal epithelia and/or endothelia. Arrows from the membrane-bound GRP94 show the stabilization of TJs and AJs by a yet-to-be-defined mechanism. (**B**) The diagram depicts the effects of different chaperones on AJ/TJ disassembly in activated/stressed epithelial and endothelial cells. Both chaperone-dependent inhibition and acceleration of junctional disassembly are shown.

**Figure 3 cells-13-00370-f003:**
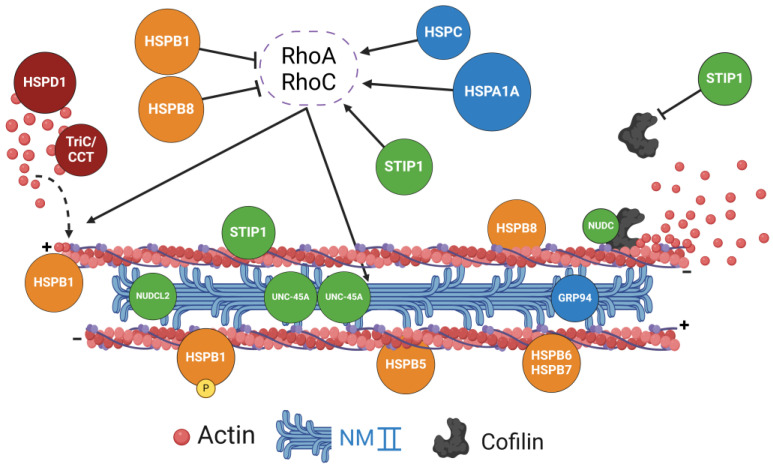
Regulation of the actomyosin cytoskeleton by different chaperones and co-chaperones. The diagram presents the reported interactions of HSP chaperones and co-chaperones with actin, non-muscle myosin II (NM II) motor, and actin-depolymerizing protein, cofilin. It also shows the chaperon-dependent activation and inhibition of Rho GTPases.

**Table 1 cells-13-00370-t001:** Regulation of epithelial and endothelial junctions by HSPA/Hsp70 chaperones.

HSPA/Hsp70 Modulation	Experimental System	Effects on Intercellular Junctions	Reference
HSPA1A/Hsp72 siRNA knockdown	Lung (A549) and mammary (MCF-7) epithelial cells	Decreased expression of E-cadherin and occludin	[40]
HSPA8/Hsc70 siRNA knockdown	NRK-52E kidney epithelial cells	Decreased expression and junctional accumulation of E-cadherin	[41]
HSPA1A/Hsp72 overexpression	TGF-β-stimulated NRK-52E cells	Attenuated TGF-β-induced decrease in E-cadherin expression	[42]
Induction of Hsp70s by geranylgeranylacetone (GGA)	TGF-β-stimulated A549 cells	Attenuated TGF-β-induced decrease in E-cadherin expression	[43]
HSPA1A/Hsp72 antisense knockdown	Monochloramine-exposed Caco-2 colonic epithelial cells	Promoted oxidant-induced decrease in TEER and an increase in the mannitol flux	[44]
HSPA1A/Hsp72 antisense knockdown	*C. difficile* toxin A-exposed Caco-2 cells	Promoted oxidant-induced decrease in TEER and increase in the mannitol flux	[45]
HSPA1A/Hsp72 overexpression	Gliadin-exposed Caco-2 cells	Attenuated gliadin-induced AJ disassembly	[46]
HSPA12B overexpression	LPS-treated human umbilical vein endothelial cells (HUVEC)	Attenuated LPS-induced FITC–dextran permeability; restored VE-cadherin expression	[47,48]
Induction of Hsp70s by TRC051384	LPS-treated HUVEC	Attenuated LPS-induced FITC–dextran permeability and restored the decreased E-cadherin, occludin and ZO-1 expression	[49]
Combined HSPA1A/Hsp72 and HSPA2/Hsp70-3 knockout in mice	DSS colitis	Promoted DSS-induced gut barrier breakdown and decreased junctional accumulation of ZO-1	[25]
Induction of Hsp70s by GGA in rats	Obstructive nephropathy	Restored E-cadherin expression in the obstructed kidney	[42]
Intratracheal administration of HSPA12B siRNA in mice	Cecal ligation and puncture model of sepsis	Increased vascular permeability in septic lungs	[47]
Transgenic mice with human HSPA12B overexpression	Myocardial injury and reperfusion	Attenuated vascular leakage; increased ZO-1 expression	[50]

**Table 2 cells-13-00370-t002:** Effects of HSPC/Hsp90 inhibition on intercellular junctions and tissue barriers.

Hsp90 Inhibitor	Experimental System	Effects on Intercellular Junctions and Cytoskeleton	Reference
Radicicol	LPS-stimulated bovine pulmonary arterial endothelial cells (BPAEC)	Prevented LPS-induced TEER decrease; attenuated AJ disassembly; restored VE-cadherin levels	[65]
17-AAG	LPS-stimulated human lung microvascular endothelial cells (HLMVEC)	Prevented LPS-induced TEER decrease; suppressed MLC phosphorylation	[66]
Radicicol, 17-AAG, or 17-DMAG	TGF-β-stimulated BPAEC	Prevented TGF-β-induced TEER decrease; attenuated AJ disassembly	[67]
AUY-922	Hydrochloric acid-exposed HLMVEC	Attenuated acid-induced TEER decrease, AJ disassembly, and F-actin remodeling	[68]
AT 13387 or AUY-922	SARS-CoV2 spike protein-stimulated HLMVEC	Attenuated spike protein-induced TEER decrease, AJ disassembly, and F-actin remodeling	[69]
17-AAG	LPS-induced uvetitis in rats	Attenuated LPS-induced retinal vascular permeability; restored ZO-1 and occludin expression	[70]
17-DMAG	Intracerebral hemorrhage in rats	Attenuated increased permeability of brain vasculature; partially restored ZO-1, occludin, and claudin-5 expression	[71]
17-DMAG	Cerebral ischemic stroke in mice	Attenuated increased permeability of brain vasculature; partially restored ZO-1 and occludin expression	[72]
17-DMAG	Traumatic brain injury in mice	Attenuated increased permeability of brain vasculature; partially restored ZO-1 and occludin levels	[73]
17-AAG	Human corneal epithelial cells	Promoted AJ assembly	[74]
Geldanamycin	Colon cancer cells (HCT116)	Promoted AJ assembly	[75]
Ganetespib	HCT116 and HT-29 cells	Increased E-cadherin expression	[76]
Geldanamycin	Gα12-overexpressing MDCK cells	Attenuated Gα12-induced TEER decrease and TJ disassembly	[77]
shRNA-mediated knockdown of GRP94	HeLa cells	Increased MLC phosphorylation and assembly of actomyosin stress fibers	[78]
Gp93 mutation that abrogates its tissue expression	*Drosophila*	Abnormal assembly of septate junctions in the intestinal epithelium	[79]
Inducible whole-body GRP94 knockout	Mice	Epithelial TJ disassembly in the ileum; bacterial translocation from the gut	[80]
Hepatocyte-specific GRP94 knockout	Mice	Disruption of AJs and gap junctions	[81]

**Table 3 cells-13-00370-t003:** Clinical and experimental evidence of UNC-45A-dependent regulation of epithelial barriers.

Loss of UNC-45A Function	Human Disease or Experimental Condition	Symptoms or Biological Effects	References
UNC-45A mutants decreasing its expression	Osteo-oto-hepatic-enteric syndrome	Diarrhea, cholestasis, and impaired hearing	[133,134,135]
UNC-45A mutants decreasing its expression	Aagenaes syndrome	Cholestasis and lymphedema	[136]
UNC-45A mutants decreasing its expression	Congenital diarrhea disorder	Diarrhea, duodenal villous atrophy, erosion of apical microvilli	[137]
Chemically induced UNC-45A mutation	Zebrafish	Apical vacuolar inclusions, microvilli shortening	[137]
UNC-45 mutants decreasing its expression	*Drosophila*	Increased intestinal permeability	[138]
CRISPR/Cas9-mediated knockout of UNC-45A	SK-CO15 and HT-29 human colonic epithelial cells	Increased barrier permeability, AJ and TJ disassembly, disorganization of the perijunctional actomyosin belt	[138]

**Table 4 cells-13-00370-t004:** HSPB-dependent regulation of epithelial and endothelial junctions.

HSPB Modulation	Experimental System	Effects on Intercellular Junctions and Cytoskeleton	References
HSPB1/Hsp27 knockdown and overexpression	LNCaP prostate cancer cells	Hsp27 overexpression disrupted AJs and TJs and downregulated E-cadherin levels; Hsp27 depletion promoted AJ/TJ assembly and E-cadherin expression	[157]
HSPB1/Hsp27 knockdown and overexpression	HCT116 and HT-29 colon cancer cells	Hsp27 overexpression downregulated E-cadherin levels; Hsp27 depletion increased E-cadherin expression	[156]
HSPB1/Hsp27 knockdown	ASPC-1 and PANC-1 pancreatic cancer cells	Increased E-cadherin and β-catenin expression	[155]
HSPB1/Hsp27 overexpression	Normal or TGF-β-treated NRK-52E cells	Increased E-cadherin expression; prevented TGFβ-induced loss of E-cadherin	[158]
HSPB1/Hsp27 knockdown and overexpression	ATP-depleted mouse or human kidney tubular epithelial cells	Hsp27 knockdown promoted metabolic stress-induced disruption of the paracellular barrier, AJ disassembly, and remodeling of perijunctional F-actin. Hsp27 overexpression attenuated barrier disruption, AJ and cytoskeletal remodeling in metabolically stressed cells	[159]
HSPB1/Hsp27 overexpression	Clostridial Toxin B exposed Caco-2 cells	Attenuated Toxin B-induced decrease in TEER	[160]
HSPB1/Hsp27 overexpression	Oxygen-glucose deprivation of HBMEC	Attenuated ischemia-induced increase in endothelial permeability and AJ disassembly	[161]
Overexpression of HSPB1/Hsp27 phosphomimetic	Hypoxia or TGF-β treatment of pulmonary artery microvascular endothelial cells	Prevented increase in permeability and endothelial gap formation induced by hypoxia or TGF-β	[162]
HSPB1/Hsp27 knockdown	Control or thrombin-activated HDMEC	No effect on normal endothelial permeability; enhanced thrombin-induced disruption of the endothelial barrier	[163]
Pharmacological inhibition of Hsp27 oligomerization	Thrombin-activated HUVEC	Exaggerated thrombin-induced disruption of endothelial barrier	[163]
HSPB1/Hsp27 overexpression in mice	Hepatic ischemia and reperfusion	Suppressed the increased renal vascular permeability	[164]
HSPB1/Hsp27 overexpression in mice	Hepatic ischemia and reperfusion	Suppressed the increased hepatic vascular permeability and attenuated the loss of bile canalicular F-actin	[165]
HSPB1/Hsp27 overexpression in mice	Brain ischemia and reperfusion	Attenuated increased brain vascular permeability and preserved VE-cadherin and occludin expression	[166]
Endothelial-specific overexpression of HSPB1/Hsp27 in mice	Brain ischemia and reperfusion	Attenuated increased brain vascular permeability	[161]
Renal tubular epithelial cell overexpression of HSPB1/Hsp27 in mice	Unilateral urethral obstruction	Attenuated AJ disassembly in the tubular epithelium	[167]
HSPB5/αB-crystallin overexpression	HONE1-2 nasopharyngeal carcinoma cells	Induced AJ assembly and increased E-cadherin expression	[168]
HSPB5/αB-crystallin knockdown	SW480 and LoVo colon cancer cells	Increased E-cadherin expression	[169,170]
Systemic administration of the cell-permeable recombinant HSPB5	DSS colitis model	Inhibits DSS-induced increase in gut permeability; restores ZO-1 and E-cadherin expression in the inflamed colonic mucosa	[171]
HSPB8/Hsp22 overexpression	Oxygen/glucose deprivation in bEnd.3 cells	Attenuated endothelial barrier disruption and F-actin polymerization triggered by the ischemia model	[172]
Lentiviral delivery of HSPB8/Hsp22 into the mouse brain	Brain ischemia and reperfusion	Attenuated increased brain vascular permeability and preserved claudin-5 and occludin expression	[173]
Lentiviral delivery of HSPB8/Hsp22 into the mouse brain	Intracerebral hemorrhage model	Attenuated increased brain vascular permeability and preserved claudin-5 and occludin expression	[174]

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
