# Peer review of "Regulation of Epithelial and Endothelial Barriers by Molecular Chaperones"

_cells, 2024, doi:10.3390/cells13050370_

Round 1

Reviewer 1 Report

Comments and Suggestions for Authors

The authors provide an interesting and very comprehensive review on the role of chaperones in regulation of epithelial and endothelial cellular junctions. They adress an emerging field in research with great importance which is far from beeing understood. The manuscript is clearly structured and excellently written. The figures are comprehensible and an adequate outlook is provided.

I have no issues and recommend publication of that manuscript in its present form.

Author Response

We thank the reviewer for very positive evaluation of our paper and suggestion to accept the manuscript without revision.

Reviewer 2 Report

Comments and Suggestions for Authors

In this manuscript the main question addressed is the regulation of epithelial and endothelial barriers by molecular chaperones. The authors focus on heat shock protein chaperones, their co-chaperones, and chaperonins discussing the roles of chaperones in the regulation of the steady-state integrity of epithelial and vascular barriers as well as disruption of these barriers by pathogenic factors and extracellular stressor. Moreover, they discuss the chaperone-assisted assembly of the actomyosin cytoskeleton since cytoskeletal is essential for the junctional integrity.

In this review, the authors discuss accumulating data regarding the roles of molecular chaperones in regulating intercellular junctions and barrier properties of epithelial and endothelial layers in vitro and in vivo. The discussed proteins are Hsp70, Hsp90, Hsp110, Hsp90, GRP94 and their co-chaperones. In my opinion it is interesting the discussion of the co-chaperone function related to intercellular junctions and barrier properties with clinical and experimental evidence since it can be relevant for feature study in the field. The paper address the need to build a comprehensive ‘junctional chaperome’ and determine the interactions and functional interplay between key chaperones and chaperonins composing this molecular network. Moreover, it is important to resolve the controversial role of HSPA/Hsp70s and HSPC/Hsp90s.  

The review reports data available in literature and compared with other published material does not add novelty.

The conclusions are consistent with the references discussed and the authors provided a critical comment of current data with the purpose to stimulate the scientific community to undertake new experiment to better understand the role of molecular chaperones in regulating intercellular junctions and barrier properties of epithelial and endothelial layers since this will be useful for the discovery of novel mechanisms that regulate the disruption and repair of epithelial and endothelial barriers during tissue injury and inflammation.

The references are appropriate.

Figure are of good quality and are well integrated in the text. The use of table was a good choice to better understand the text with the possibility with a rapid access to the references.    

In conclusion, I suggest to accept the manuscript in the present form, it contains table and figures useful for the scientific community of the field. The authors declare clearly the scope of the paper, it is just a discussion accumulating data regarding the role of molecular chaperones in regulating intercellular junctions and barrier properties of epithelial and endothelial layers. Even if the manuscript does not include novelty or particular new hypothesis it would be useful to draw attention to the topic especially for the pathological implications.

Author Response

(The authors gave the same response as above.)

Reviewer 3 Report

Comments and Suggestions for Authors

The paper is a very detailed review mostly on Heat shock proteins.  Therefore I suggest to modify the title more related to what is described. 

Comments on the Quality of English Language

Apart from a repetition of prepositions (such as since in line 104 and 108) the paper is very well written with Tables and Figures.

Author Response

We thank the reviewer for the positive evaluation of our manuscript and for valuable comments that are addressed below.

Reviewer Comment 1: The paper is a very detailed review mostly on Heat shock proteins.  Therefore, I suggest to modify the title more related to what is described. 

Response: We appreciate this comment, but we would prefer to keep the current manuscript tile for the following reasons. First, while the vast majority of molecular chaperons are assigned to different families of heat shock proteins, this seems to be for a nomenclature convenience, since large number of these proteins are not inducible by heat chock or other environmental stressors. Second, this review discusses several important co-chaperones, such as NUDC proteins and UNC-45A, that do not formally belong to HSPs. Furthermore, the  TRiC/CCT chaperonins, while recently being included into the HSP superfamily, are better known as a standing alone chaperonin complex. 

We believe that having a broader, more inclusive title will attract more readers attention to the present paper.

Reviewer comment 2: Apart from a repetition of prepositions (such as since in line 104 and 108) the paper is very well written with Tables and Figures.

Response: We appreciate this comment and made some editing in the revised manuscript.

Reviewer 4 Report

Comments and Suggestions for Authors

This review article discusses the roles of heat-shock proteins in regulation of the barrier function, especially in regulation of tight and adherens junction stability. The review is well-written, comprehensive, includes several informative schematics and tables and nicely discusses current knowledge on the topic. Overall, it is a significant and timely contribution to the field. The only comment of this reviewer is that the authors should try to better clarify, differentiate and discuss the roles of the inducible vs the constitutively expressed HSPs, especially within the HSP70 family; this is important, since the inducible ones are expressed in stress conditions or in cancer and this can be informative regarding the status of tight and adherens junctions in these conditions.

Author Response

We thank the reviewer for the positive evaluation of our manuscript and for a valuable comment that is addressed below.

Reviewer Comment: the authors should try to better clarify, differentiate and discuss the roles of the inducible vs the constitutively expressed HSPs, especially within the HSP70 family.

Response: We appreciate this important comment, however the available data do not allow to conclude whether the inducible and non-inducible HSPs may have different functions at tissue barriers. Only a few selective members of different HSP families were investigated and there no studies aimed at back-to-back evaluation of functional effects on inducible versus non-inducible HSPs under similar experimental conditions. For the Hsp70 family, only three members (HAPA1A, HSPA8 and HSP12B) have been investigated and all appear to have barrier-protective functions.

Reviewer 5 Report

Comments and Suggestions for Authors

In the present manuscript Lechuga et al. intended to review the specific role of chaperons protein in stabilizing/destabilizing cell/cell contacts of epithelial or endothelial barriers.

Their work was developed in an extensive approach giving a pretty unique review in the field.

There are few suggestions/comments:

-the white characters of both Fig.1 and 2 are not very clear, they may be fixed using the bold option . In Fig.3 the legend for NM II is missed. Furthermore, it is suggested to explain what cofillin is as reported in lane 512.

- Pag.6, lane165: please the cell lines need to be better classify: both A549 and MCF-7 are from carcinomas and not normal epithelia cells.

- Conclusions: it is suggested to include here a paragraph about the relevance of the role of this class of proteins in particular pathologists like infection, inflammation-related diseases (colon!!) and cancer leaving as it is the last conclusive paragraph.

Author Response

We thank the reviewer for the positive evaluation of our manuscript and for valuable comments that are addressed below.

Reviewer comment 1: the white characters of both Fig.1 and 2 are not very clear, they may be fixed using the bold option . In Fig.3 the legend for NM II is missed. Furthermore, it is suggested to explain what cofillin is as reported in lane 512.

Response: We appreciate these comments and modified all three Figures and the Figure 3 legend as suggested by the reviewer.

Reviewer Comment 2:  Pag.6, lane165: please the cell lines need to be better classify: both A549 and MCF-7 are from carcinomas and not normal epithelia cells.

Response: Thanks for this comment. We have highlighted in different parts of the revised papers that many used epithelial cell lines are derived from different solid tumors and are not norma epithelial cells.

Reviewer Comment 3: To expand the conclusion by adding a paragraph about the relevance of the role of this class of proteins in particular pathologists like infection, inflammation-related diseases (colon!!) and cancer.

Response: We appreciate this comment. We repeatedly mentioned in different places of the manuscript that many HSPs are upregulated in various cancers and play roles in tumorigenesis. Including this statement in the Conclusion would be repetitive. Much less is known about functional roles of different chaperones in tissue inflammation and infection, and in the Conclusion, we included the focus on these diseases as an important line of future investigations. Further expanding this part would not provide additional new information or new focus for this already long manuscript.

Reviewer 6 Report

Comments and Suggestions for Authors

This is a well organized review article that focuses on the network of the regulation of intercellular adhesions, and the systematic analysis of chaperone functions at epithelial and endothelial barriers.

I think his review has completely described the functions and mechanisms of heat shock protein chaperones  that assisted regulation of intercellular junctions.  It would be an important article that as a hallmark of chaperone-mediated control of tissue barriers.

Author Response

(The authors gave the same response as above.)
